# Characterization of Wildfires and Harvesting Forest Disturbances and Recovery Using Landsat Time Series: A Case Study in Mediterranean Forests in Central Italy

Carmelo Bonannella [1,2,3], Gherardo Chirici [3,4], Davide Travaglini [3,4], Matteo Pecchi [3], Elia Vangi [3], Giovanni D'Amico [3] and Francesca Giannetti [3,4,*]

1   OpenGeoHub Foundation, 6708 PW Wageningen, The Netherlands; carmelo.bonannella@opengeohub.org
2   Laboratory of Geo-Information Science and Remote Sensing, Wageningen University & Research, 6700 AA Wageningen, The Netherlands
3   Geolab Laboratory of Forest Geomatics, Department of Agriculture Food Environment and Forestry, University of Florence, 50145 Florence, Italy; gherardo.chirici@unifi.it (G.C.); davide.travaglini@unifi.it (D.T.); matteo.pecchi@unifi.it (M.P.); elia.vangi@unifi.it (E.V.); giovanni.damico@unifi.it (G.D.)
4   ForTech Laboratorio Congiunto, University of Florence, 50145 Florence, Italy
*   Correspondence: francesca.giannetti@unifi.it

**Abstract:** Large-scale forest monitoring benefits greatly from change detection analysis based on remote sensing data because it enables characterizing forest dynamics of disturbance and recovery by detecting both gradual and abrupt changes on Earth's surface. In this study, two of the main disturbances occurring in Mediterranean forests, harvesting operations and forest fires, were analyzed through the analysis of Landsat Times Series images in a case study in Central Italy (Tuscany region). Disturbances were characterized based on their distinct temporal behaviors before and after the event: a period of 20 years (1999–2018) was used to extract and analyze at pixel level spectral trajectories for each disturbance and produce descriptive temporal trends of the phenomena. Recovery metrics were used to characterize both short- (5 years) and long-term aspects of recovery for harvested and burned areas. Spectral, recovery, and trend analysis metrics were then used with the Random Forest classifier to differentiate between the two disturbance classes and to investigate their potential as predictors. Among spectral bands, the Landsat SWIR 1 band proved the best to detect areas interested by harvesting, while forest fires were better detected by the SWIR 2 band; among spectral indices, the NBR scored as the best for both classes. On average, harvested areas recovered faster in both short- and long-term aspects and showed less variability in the magnitude of the disturbance event and recovery rate over time. This tendency is confirmed by the results of the classifier, which obtained an overall accuracy of 98.6%, and identified the mean of the post-disturbance values of the trend as the best predictor to differentiate between disturbances.

**Keywords:** wildfire; harvest; forests; forest fires; coppices; classification of forest disturbances; time series

## 1. Introduction

The Mediterranean Basin represents one of the five different regions that compose the Mediterranean eco-region area, with other regions being: California, southwestern and southern Australia, the Western Cape Region in South Africa, and central Chile [1]. Among these regions, the Mediterranean Basin is the biggest [2]. It is also estimated that this area contains more than 10% of the world's vascular plant biodiversity with about 290 different indigenous taxa [2–5].

The Mediterranean region is historically subject to significant human pressures that have determined a profound transformation of the natural landscape [4]. In addition, during the last two decades, this pressure is further increased due to the action of the climate

change phenomenon that caused an alteration in the frequency and intensity of disturbance events [6–10] and a possible modification of the traditionally human activities such as timber harvesting because climate change could alter species composition, physiology, and regrowth [10–12].

In Mediterranean areas, and Italy is not an exception, fire represents a common and important historical natural disturbance agent for Mediterranean forests. Recent statistics realized by the European Forest Fire Information System (EFFIS) showed that, during the last year, the wildfire activities increased in all Mediterranean countries, especially in Italy, where, just in 2021, a series of catastrophic events in the Sardinia, Sicily, and Calabria regions was registered [13–15]. The increasing of wildfire activities in Mediterranean forest ecosystems, despite forest species having adaptation mechanisms such as resprouting capacity, seed bank persistence, and better dispersal capacity of seeds [13–15] to survive wildfire events, can compromise the stability of slopes and the regeneration rate of forests [16–18].

The other typical historical disturbance of Mediterranean forests is harvesting [19,20]. In Italy, clearcut of coppices forest is the most typical forest harvesting activity [20–22] and the most typical human forest disturbance, since coppices represent the main forest management regime. In coppice forests, regrowth after the cut occurs thanks to the rapid asexual regeneration and the sprouting of new shoots from the stump [23]. However, due to climate change, it is important to monitor the regrowth rate of harvested forests, to reduce potential degradation. Therefore, monitoring these two different types of Mediterranean forests disturbances, their effects, and differences in recovery rates is today fundamental to support forest strategies at a national and international scale in the context of sustainable forest management [3,24], biodiversity conservation [25], and carbon sink balance [6].

Optical remote sensing data are considered one of the most efficient tools to map and monitor changes on a regular and continuous basis at different spatial scales, from the global/national [12,20,26] to the regional/local [10,22,27] level. The use of optical remote sensing tools is very common, mainly for six reasons, which are: (i) they provide a complete painting of the study area, (ii) they are always available, (iii) they have a high degree of homogeneity and there are not any influence of human actions, (iv) the images are in digital format and easily integrated with other spatial data, (v) they are available at a low price, (vi) there is an increasing trend in the production of data [28].

With about 50 years of data, the Landsat series represents the main data source for large-scale monitoring programs. This aspect, coupled with the free data policy, the high spatial resolution (30 m), and the low temporal resolution (about 16–18 days), has allowed the development of temporally dense pixel-level analysis on a large scale and across different thematic domains [29–32].

Change detection analysis allows the detection and assessment of changes occurred in a specific area by comparing images of the same area with different acquisition dates. Change detection can be conducted using a bi-temporal approach, where only two images are compared (for forest disturbances, an image before and after the disturbance event are used) or a time series approach [33]. The time series approach enables characterizing the temporal forest dynamics of disturbance and recovery by detecting both gradual and abrupt changes. In this case, the spectral trajectory of a given pixel can be interpreted as an ecological response curve whose shape reveals information about the underlying process of change [21,34,35].

TS methods can be divided into four different categories: threshold-based, curve fitting, trajectory fitting, and finally trajectory segmentation [33]. The last two approaches in particular are widely used to monitor forest disturbance [36,37]. These approaches use a dense series of images to detect a characteristic spectro-temporal signature [33,38]. Trends of forested pixels over time are described by a curve characteristic of the phenomenon occurred: trajectory fitting methods use the end year of the disturbance interval, the pre-disturbance mean reflectance, and the post-disturbance mean as parameters to build the spectral signature, with the slope representing an estimation of the rate at which degrada-

tion or recovery phenomena occur [39,40]. Slope positive or negative values indicate an increase or decrease in vegetation, respectively. In this sense, Kennedy et al. [38] hypothesized four model or *disturbance classes* that, fitted to observed time series at pixel level, could describe long term behavior of the spectral trajectory and the phenomenon occurred:

- simple disturbance;
- disturbance followed by re-vegetation;
- ongoing re-vegetation from a disturbance event occurred before the time period analyzed;
- re-vegetation from prior disturbance to a stable state reached during the observation period.

Following these premises, the main objective of this study was to investigate both short- (5 years) and long-term (>5 years) forests spectro-temporal responses to different types of disturbance in Mediterranean forests and characterize the behavior of such forest disturbance. The focus is on two types of disturbance: clearcut and wildfires.

These disturbance events were analyzed using a twenty-year (from 1999 to 2018) Landsat time series (LTS) composite, realized using one image per year, and two reference datasets (i.e., clearcut polygons and wildfire polygons for which the years of disturbances were known). Firstly, temporal spectral trajectories based on time series Landsat bands and vegetation indices were extracted at a pixel level, fitted and aggregated to produce a single trajectory for each type of disturbance, and then compared. Secondly, disturbances and recovery rates of a forest were characterized using existing metrics of short and long term recovery, already tested in Canadian boreal forests by White et al. [12] and in Mediterranean forests by Chirici et al. [35] with the spectral signature analyzed using spectral and trend analysis metrics. Due to the diversity of boreal and Mediterranean forests conditions, the study was conducted assuming that both disturbances and recovery phenomena in Mediterranean forests evolve differently than in the boreal forests. Therefore, information derived from the trends and the metrics were combined with the attempt to answer the following questions:

1. Which is the most effective spectral variable regrowth trajectory to detect disturbances and recovery effects in the Mediterranean forests?
2. Are there any differences in the spectral trends and recovery conditions among the two classes of disturbances (i.e., clearcut and wildfire) captured by LTS analysis and all derived metrics? Can these differences be used to obtain a distinct profile for each disturbance?

## 2. Materials and Methods

### 2.1. Study Area

The study was conducted in the forested area covered by a single Landsat scene (Path: 192, Row: 030); the area covered falls almost entirely in the administrative region of Tuscany, Central Italy (Figure 1).

It is mainly a hilly region (mean elevation = 320 m above sea level), characterized by large altitude differences (from sea level up to 1900 m in the topmost part) and moderate slopes (mean slope = 11%). The coast line and the immediate areas behind it have a Mediterranean climate [41], which results in a mixed sclerophyll forest dominated by Holm oak (*Quercus ilex* L.) and Cork oak (*Quercus suber* L.) with other shrubs commonly found in maquis formations; other tree species include Maritime pine (*Pinus pinaster* Aiton), Italian stone pine (*Pinus pinea* L.), and Aleppo pine (*Pinus halepensis* Mill.), present mainly as artificial plantations. These species are well adapted to low intensity wildfires: the Aleppo pine and the Maritime pine have fire-activated seed banks that germinate, grow, and mature rapidly following a fire in order to reproduce and renew the seed bank before the next fire, while the Holm oak resprouts vigorously after clearcut harvesting (coppices) and wildfire disturbances [15,42].

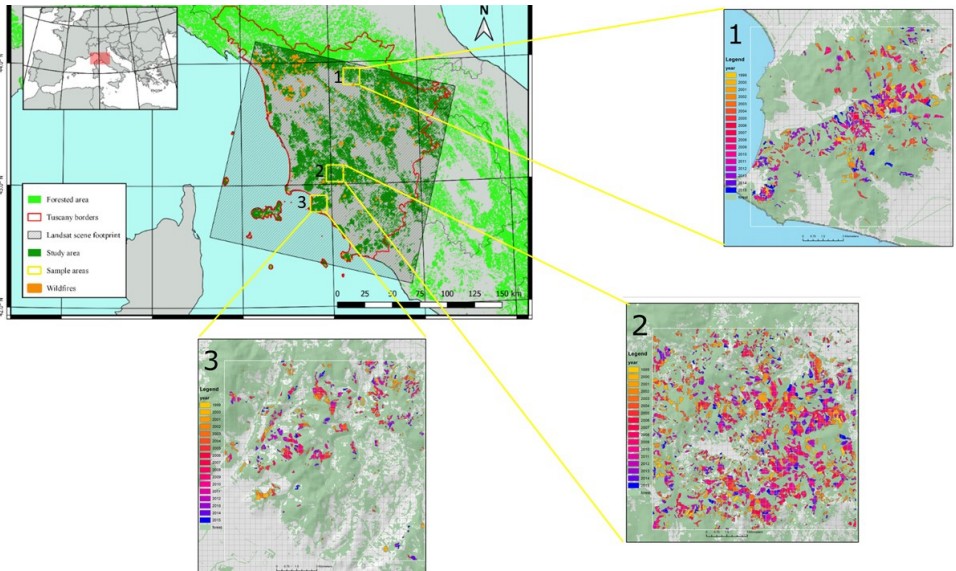

**Figure 1.** Localization of the study area in Central Italy. The shaded area shows the footprint of the Landsat WRS-2 scene used, while in the subfigures (**1**–**3**) are reported the clearcut coppices area by years of the sample areas.

Moving inland, the climate shifts to temperate oceanic [41] and the vegetation changes towards a mixed temperate forest dominated by Turkey oak (*Quercus cerris* L.), Downy oak (*Quercus pubescens* L.), and Sweet chestnut (*Castanea sativa* L.) with other tree species like European beech (*Fagus sylvatica* L.) and European hop-hornbeam (*Ostrya carpinifolia* Scop.); Black pine (*Pinus nigra* J.F. Arnold) is present mainly as a species used for artificial plantations. These broadleaves species are less adapted to wildfire compared to the previous ones; however, they are largely used for firewood production and intensively managed as coppices, while Black pine plantations are mostly unmanaged and are characterized by a large amount of accumulated flammable organic components, which represent a potentially large fire risk [43].

Six out of the 14 European Forest Types [44,45] are represented in the study area.

*2.2. Landsat Time Series Data*

For this study, we enlarged the LTS used by Chirici et al. [35], spanning the period 1999–2018. The LTS used is composed by one image per year with cloud cover <5% acquired by Landsat 5 TM, Landsat 7 ETM+, and Landsat 8 OLI (Table A1). The images were downloaded from the USGS web service https://earthexplorer.usgs.gov, access on-line the 15 June 2021, (Table A1) and were pre-processed using the same methodology of Chirici et al. [35]. The images used were acquired during the summer months due to the correspondence with the growing season for most forest species in the study area [46] and to avoid phenological differences. For each image, we used the information of six Landsat bands (Blue, Green, Red, NIR, SWIR 1, and SWIR 2) and we calculated seven different spectral indices: Normalized Difference Vegetation Index (NDVI) [47], Enhanced Vegetation Index (EVI) [48], Soil Adjusted Vegetation Index (SAVI) [49], Modified Solid Adjusted Vegetation Index (MSAVI) [50], Normalized Burned Ratio (NBR) [51] and Normalized Burned Ratio 2 (NBR2) [52], and Normalized Difference Moisture Index (NDMI) [53] (Table A2).

*2.3. Forest Types Classes*

To focus the spectral extraction only on forested areas and to compare the disturbances among different forest types, we used two cartographic layers that cover the whole study area: (i) the National Forest mask [54] and (ii) the Tuscany Regional Forest Inventory (TRFI)

dataset. The forest mask was used to derive information of undisturbed forest, while from the TRFI we extracted the information related to forest types [55].

### 2.4. Disturbances Reference Geodatabase

Three reference spatial polygons geodatabases, available in Tuscany, were used to compare the differences in the spectral trajectory and recovery rates over time between disturbed (i.e., wildfire and harvest) and undisturbed forest areas (Figure 2). In details, we used: (i) the geodatabase provided by Carabinieri Forestali for wildfire areas (i.e., 2005–2015) [56], which consists of a total of 3394 ha of mapped forest fire areas; (ii) the updated version of the harvesting reference geodatabase (i.e., 1999–2018) provided by Chirici et al. [35] for three sample quadrats (Figure 2) for a total of 8545 ha of harvested area, and (iii) a reference geodatabase of undisturbed forests (i.e., 1999–2018) visually interpreted for the present study by means of high-resolution orthophotos and Landsat images (1999–2018) for a total of 2902 ha of undisturbed forest. In Table 1, we report the number of pixels for each class (i.e., wildfires disturbed forest area, harvesting disturbed forest areas, undisturbed forest area).

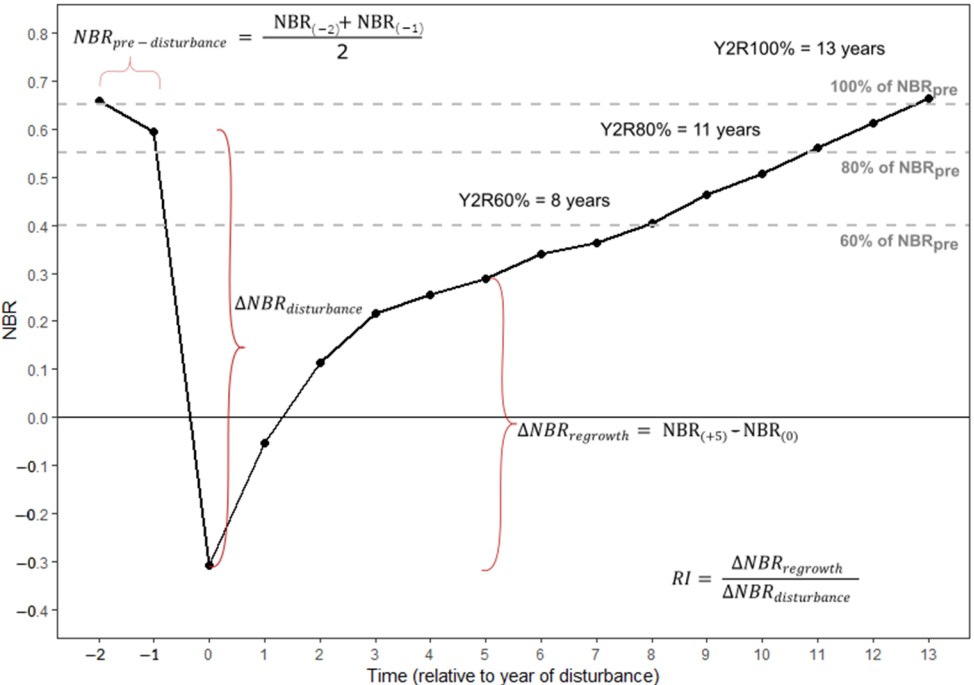

**Figure 2.** Description of all NBR recovery metrics.

**Table 1.** Frequency of the pixels before and after the correction process, by disturbance and forest type.

| Forest Type | Undisturbed Areas | | Harvesting | | Wildfires | | Total | |
|---|---|---|---|---|---|---|---|---|
| | Pre-correction | Post-correction | Pre-correction | Post-correction | Pre-correction | Post-correction | Pre-correction | Post-correction |
| *Abies alba* (Silver fir) | 0 | 0 | 18 | 0 | 0 | 0 | 18 | 0 |
| *Alnus glutinosa* (Common alder) | 224 | 0 | 277 | 229 | 26 | 0 | 527 | 229 |
| *Castanea sativa* (Sweet chestnut) | 2289 | 1372 | 4904 | 2962 | 6254 | 1348 | 13,447 | 5682 |
| *Cupressus sempervirens* (Mediterranean cypress) | 554 | 334 | 178 | 75 | 252 | 171 | 984 | 580 |
| *Fagus sylvatica* (European beech) | 0 | 0 | 53 | 0 | 3006 | 0 | 3059 | 0 |
| Maquis formations | 485 | 0 | 1262 | 807 | 1971 | 1002 | 3718 | 1809 |
| Mixed plantations of non-native species | 0 | 0 | 478 | 338 | 78 | 0 | 556 | 338 |
| Montane shrubs (*Juniperus, Prunus, Spartium* spp.) | 701 | 508 | 2616 | 1866 | 1165 | 0 | 4482 | 2374 |

**Table 1.** *Cont.*

| Forest Type | Undisturbed Areas | | Harvesting | | Wildfires | | Total | |
|---|---|---|---|---|---|---|---|---|
| | Pre-correction | Post-correction | Pre-correction | Post-correction | Pre-correction | Post-correction | Pre-correction | Post-correction |
| *Ostrya carpinifolia* (European hop-hornbeam) | 1914 | 1530 | 6124 | 4095 | 2388 | 0 | 10,426 | 5625 |
| *Pinus nigra* (Black pine) | 178 | 136 | 223 | 96 | 133 | 10 | 534 | 242 |
| *Pinus pinaster* (Maritime pine) | 252 | 161 | 1171 | 638 | 11,225 | 5177 | 12,648 | 5976 |
| *Pinus pinea* (Stone pine) | 393 | 0 | 96 | 69 | 610 | 471 | 1099 | 540 |
| *Pseudotsuga menziesii* (Douglas fir) | 0 | 0 | 88 | 52 | 0 | 0 | 88 | 52 |
| *Quercus cerris* (Turkey oak) | 8755 | 5661 | 54,224 | 37,496 | 1798 | 628 | 64,777 | 43,785 |
| *Quercus ilex* (Holm oak) | 15,184 | 9995 | 19,492 | 11,584 | 3793 | 1087 | 38,469 | 22,666 |
| *Quercus pubescens* (Downy oak) | 947 | 631 | 2915 | 1753 | 4233 | 1735 | 8095 | 4119 |
| *Quercus suber* (Cork oak) | 190 | 0 | 846 | 552 | 47 | 28 | 1083 | 580 |
| *Robinia pseudoacacia* (Black locust) | 165 | 0 | 0 | 0 | 761 | 293 | 926 | 293 |
| **Total** | **32,231** | **20,328** | **94,965** | **62,612** | **37,740** | **11,950** | **164,936** | **94,890** |

*2.5. Spectral Trajectory Extraction and Spectral Trajectory Fitting*

Using the disturbances reference geodatabase described in Section 2.4, the spectral trajectories of each Landsat band and of the seven spectral indices were extracted at a pixel scale on a scene-by-scene basis. Outliers due to error measurements, clouds, haze, or other atmospheric effects were detected based on the assumption that they behave as punctual and ephemeral values (drops or spikes) in a given spectral trajectory [57]. Observations were classified as outliers if they exceeded a certain standard deviation threshold, instead of empirical fixed thresholds [58,59]. The threshold was estimated by forest type category, exploiting the differences in spectral signatures between species. Pixels were first split into different samples based on disturbance and forest type, followed by year of the image for undisturbed pixels and temporal distance for the disturbed ones.

Given the fixed response of a species to each wavelength in the same conditions (time of the year, atmospheric conditions, phenological season, water content, type of disturbance), values from each sample are inclined to converge to a central value, assuming a distribution similar enough to a normal distribution: this assumption was checked through visual inspection with Q-Q (quantile–quantile) plots, which plot the quantiles of the sample set against the quantiles of the normal distribution.

Spectral trajectories were then plotted and aggregated by disturbance and forest type to define a range of expectations that were compared with the values assumed by the standard deviation in a normal distribution; the range of expectations was established to be approximately two times the standard deviation ($2\sigma$) for all forest type classes. The arithmetic mean ($\mu$) and the standard deviation was then computed for each sample and all observations that exceeded the $\mu \pm 2\sigma$ value were classified as outliers and removed from the analysis (Table 1). These analyses allow also excluding mixed pixels (pixels on the edges of damaged areas) that usually introduce noises in spectral trajectories analysis [32].

The pixel trajectories extracted were then smoothed by constraining them to adhere to two predefined shape patterns, identified by the known ecological response of a forested pixel through time based on the phenomenon therein occurred. This approach was applied following the assumption of Kennedy et al. [34], where a pixel is classified as (i) *undisturbed* when the trajectory is characterized by a *stable state* (i.e., the reflectance values among the spectral variable of interest are that of a stable trajectory through time, with little or no fluctuation) or as (ii) *disturbed*, whether affected by low or high magnitude events, when the trajectory shows step changes in reflectance, corresponding to the disturbance event, followed in the next years by a slow recovery of the trajectory to pre-disturbance values). The R package *ShapeSelectForest* [60] was used to achieve this analysis. Knowing beforehand which phenomenon occurred (i.e., fires or harvest) and the relative year of change for each trajectory, only the following two shapes were used:

*Flat:* the pixel trajectory shows a forest in a stable condition;

*Jump:* the pixel trajectory shows a forest that suffers from abrupt changes in its structure or canopy cover due to a disturbance event (harvesting, fires).

Trajectories belonging to the undisturbed class (i.e., not disturbed forest) were constrained to adhere to the "flat" shape, while the disturbance classes (i.e., clearcut and wildfire) were constrained to the "jump" shape.

The fitted spectral trajectories were then grouped by type of disturbance and band or spectral index, which resulted in 13 samples (i.e., 6 Landsat Bands and 7 Vegetation indices) for each of the 3 classes (i.e., wildfires disturbed forest area, harvesting disturbed forest area, undisturbed forest area). At the end, an average trend for each Landsat band and each vegetation index was extracted from each class.

To characterize disturbances and the vegetation recovery, we used the absolute difference between the pre-disturbance value and the value recorded for the disturbance event and the recovery trend after the disturbance [61–66].

### 2.6. Recovery NBR-Based Metrics

The spectral trajectory of a disturbed pixel can be divided into three segments, each one representing one stage of the change event process: an undisturbed or stable segment, before the disturbance; a disturbed segment, with a consistent drop in reflectance values, which goes from the last pre-disturbance year to the year of disturbance; and a recovery segment, with reflectance values slowly coming back to to pre-disturbance levels.

To characterize each type of disturbance, for illustrative purposes, and because NBR results as the most sensitive index in detecting harvester and fires disturbances, we used the fitted NBR spectral trajectories to extract a set of metrics, i.e., conditions before and after the change event using both information derived from spectral values and linear trend descriptors (Table 2, Figure 2).

**Table 2.** Set of recovery metrics that describes conditions pre- and post-disturbance for each trajectory.

| Metric | Description |
| --- | --- |
| *Mean pre-disturbance* | Arithmetic mean of spectral values before the change event |
| *Standard deviation pre-disturbance* | Standard deviation of spectral values before the change event |
| *Slope pre-disturbance* | Direction and steepness of the trajectory before the change event |
| *ΔNBR pre-disturbance* | Arithmetic mean of the first two years before the change event |
| *ΔNBR disturbance* | Or magnitude of the event, absolute change in NBR value |
| *ΔNBR regrowth* | Absolute difference between NBR values five years after the change event and NBR values of the change event |
| *Recovery Index (RI)* | ΔNBR regrowth / ΔNBR disturbance |
| *First year post-disturbance* | Spectral value recorded in the first year after the change event |
| *Mean post-disturbance* | Arithmetic mean of spectral values after the change event |
| *Standard deviation post-disturbance* | Standard deviation of spectral values after the change event |
| *Slope post-disturbance* | Direction and steepness of the trajectory after the change event |

Although there are a conspicuous number of spectral indices and derived metrics to characterize disturbances and vegetation recovery [66–68], in this study, the definition of recovery as "the initial establishment (or pulse of vegetation), as well as a more long-term, sustained regeneration of forests at a site" proposed by Johnstone et al. [69] was adopted and the recovery metrics used by White et al. [12] to characterize both longer and shorter term aspects of the recovery process were calculated.

On the basis of fitted trend trajectories, for each of the recovery metrics we calculated, as done by Chirici et al. [35], the Years to Recovery (Y2R) metric following the approach of White et al. [12,70]. The Y2R indicates the number of years required for a pixel to attain 60%, 80%, and 100% (i.e., Y2R60%, Y2R80%, Y2R100%) of its pre-disturbance fitted trend of the vegetation index value. In our study, the pre-disturbance value used to define the Y2R was calculated as the average of the fitted index values for the two years prior to disturbance, consistent with the approach applied in White et al. (2017) [12].

### 2.7. Classification Model

A Random Forest classifier [71] was used to differentiate the two disturbance classes (i.e., wildfires disturbed forest area, harvesting disturbed forest area), using the R *random-Forest* package [72]. The final dataset containing the NBR shape-constrained trajectories, and their relative metrics, was randomly split in a 70:30 ratio, with 70% of the trajectories used for training the model and the remaining 30% used as a test set, and all computed metrics mentioned in Section 2.6 were used as predictor variables, for a total of 14 features. The most important design parameters for Random Forest are the number of trees to be generated (*ntree*) and the number of features to be selected randomly for growing each tree (*mtry*): the parametrization followed the default recommended values for the *ntree* parameter, while *mtry* was set to 14. The importance of predictor variables was assessed using the mean decrease in accuracy value returned by the algorithm instead of the mean decrease in Gini coefficient because this mechanism to compute feature importance is known to be biased, inflating the importance of continuous or high-cardinality categorical variables [73]; coefficients were normalized in a 0–100 scale of relative importance score, while the out of bag (OOB) score was used to assess the performance of the model. To obtain consistent values, the model was applied 100 times and the results averaged.

### 3. Results

#### 3.1. Spectral Response of Bands and Indices

To study the differences in the spectral trajectory of the two types of disturbances, i.e., wildfires disturbed forest area, harvesting disturbed forest area, and undisturbed forest area, run charts (Figures 3 and 4) were generated using the raw and shape-smoothed values of the six Landsat bands and each of the seven spectral indexes, measured, respectively, in surface reflectance or band ratio on the y-axis, change over time, displayed on the x-axis.

The trends for the bands were almost stationary for undisturbed areas. For each class of disturbance, we can see that the reflectance values in the segment before the detection of a disturbance event were equal to the ones computed for the undisturbed areas (Figure 3), while a sudden change in reflectance can be seen in the year of disturbance: every band is sensitive to the disturbance event, with the blue band recording the least significant absolute change in mean in all classes of disturbance. The SWIR 1 band recorded the highest values for the harvesting class followed by the SWIR 2, while the opposite was found for wildfires. However, both SWIR bands in the wildfires class recorded the highest value in the trend not in the year of disturbance, but the following year ($y = 1$). The recovery segment displayed a stable trend for all bands in the harvesting class, while more fluctuations can be seen in the wildfire class.

Trends in spectral indices (Figure 4) had the same pattern found for the bands: a stationary trend for the undisturbed areas and drops in the band ratio value in the year of a disturbance. However, the recovery segments showed consistent differences depending on which spectral index is considered; the NDVI index showed values that were equal to pre-disturbance levels after only three years, confirming its behavior to saturate rapidly after the detection of a disturbance event [35,74,75], while the NBR and the NDMI were the most useful spectral indices in detecting disturbances, showing values in absolute change in mean close to each other (Table 3).

#### 3.2. Characterizing Recovery with NBR-Based Metrics

Five years after the disturbance event, 10% of the pixels belonging to the wildfires class had a value of $\Delta NBR_{regrowth} \leq 0$ in contrast with the harvesting class with only 0.02%; all remaining pixels had values of $\Delta NBR_{regrowth}$ above that threshold, which is a sign of spectral recovery.

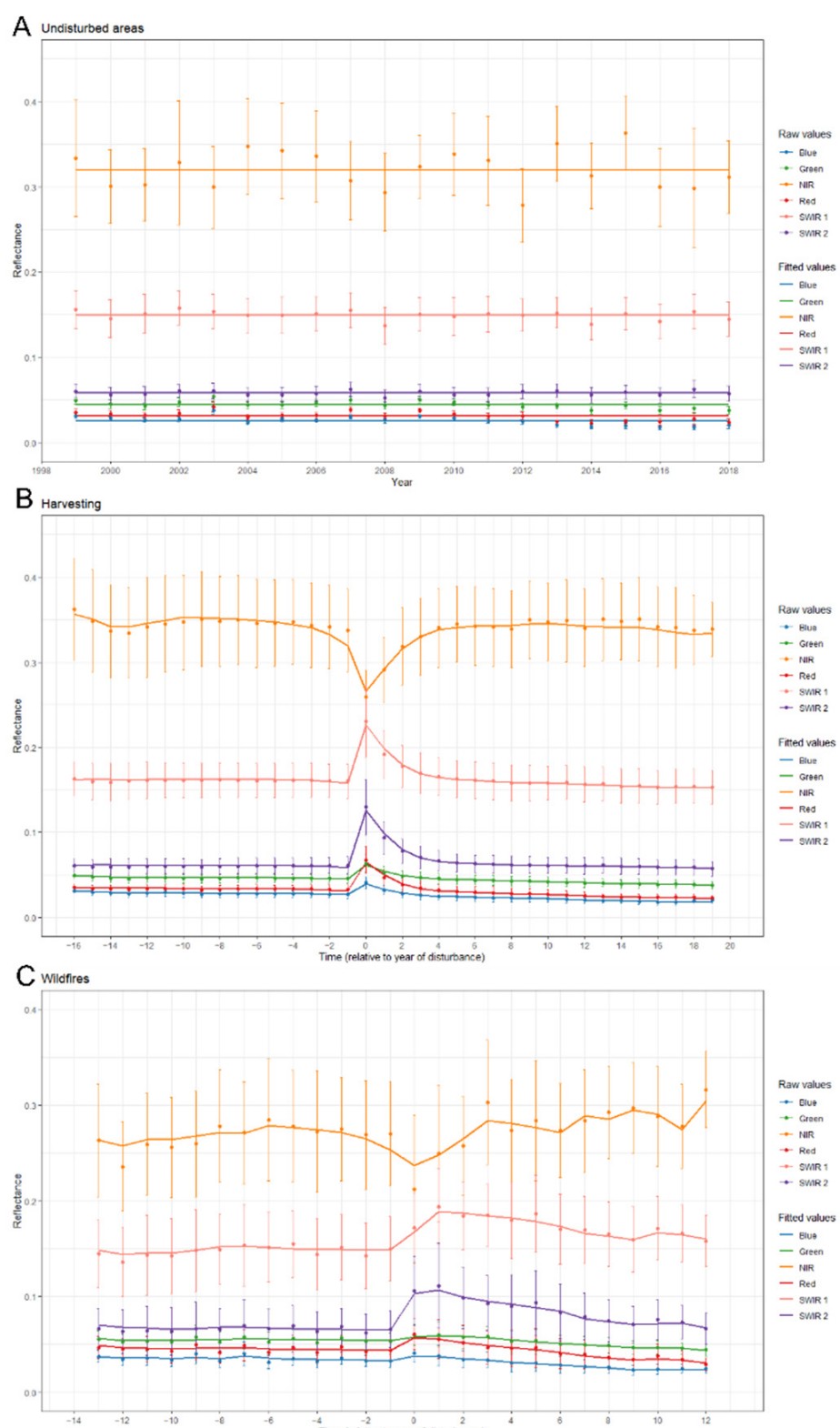

**Figure 3.** (Panel **A**)—Mean value of the bands over the 20 years analyzed for undisturbed areas. (Panel **B**)—Mean value of the bands from 16 years prior the disturbance and 19 years after for harvested areas. (Panel **C**)—Mean value of the bands from 13 years prior the fire to 12 years after. Raw values (dotted line) show the averaged temporal trend as extracted from the LTS, while fitted values (full line) show the trend without noise after the shape-smoothing algorithm. The standard deviation of the value extracted for the polygon is also reported as a vertical line.

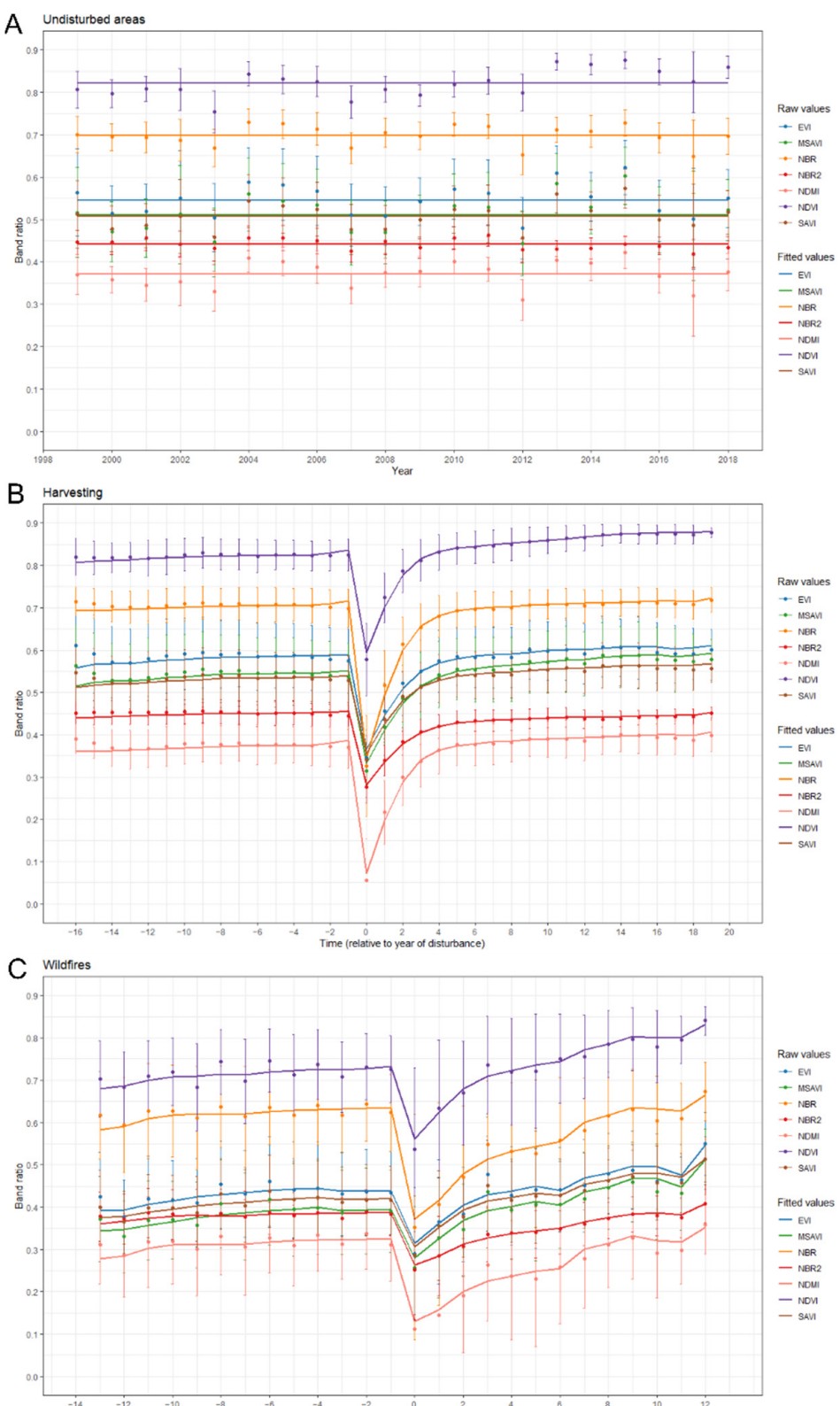

**Figure 4.** (Panel **A**)—Mean values of the spectral indices over the 20 years analyzed for undisturbed areas. (Panel **B**)—Mean value of the spectral indices from 16 years prior the disturbance and 19 years after for harvested areas. (Panel **C**)—Mean value of the spectral indices from 13 years prior the fire to 12 years after. Raw values (dotted line) show the averaged temporal trend as extracted from the LTS, while fitted values (full line) show the trend without noise after the shape-smoothing algorithm. The standard deviation of the value extracted for the polygon is also reported as a vertical line.

**Table 3.** Absolute changes in mean values by spectral band or spectral indices and disturbances; the best results for each class of disturbance are indicated in bold.

| Landsat Spectral Bands | Absolute Change in Mean | |
|---|---|---|
| | Harvesting | Wildfires |
| Blue | 0.0123 | 0.0050 |
| Green | 0.0163 | 0.0037 |
| NIR | 0.0537 | 0.0360 |
| Red | 0.0336 | 0.0132 |
| SWIR 1 | **0.0681** | 0.0184 |
| SWIR 2 | 0.0671 | **0.0384** |
| **Landsat spectral index** | | |
| EVI | 0.2245 | 0.1250 |
| MSAVI | 0.2181 | 0.1152 |
| NBR | **0.3710** | **0.2645** |
| NBR2 | 0.1746 | 0.1233 |
| NDMI | 0.3144 | 0.1947 |
| NDVI | 0.2417 | 0.1706 |
| SAVI | 0.1847 | 0.1149 |

On average, wildfires had higher absolute values and variability than harvesting, with 8% of the pixel distribution having $\Delta NBR_{regrowth} > 0.6$: the average value of $\Delta NBR_{regrowth}$ for the wildfires class was 0.245 with a standard deviation of 0.224, while for the harvesting class the average was 0.357 with a standard deviation of 0.118 (Figure 5).

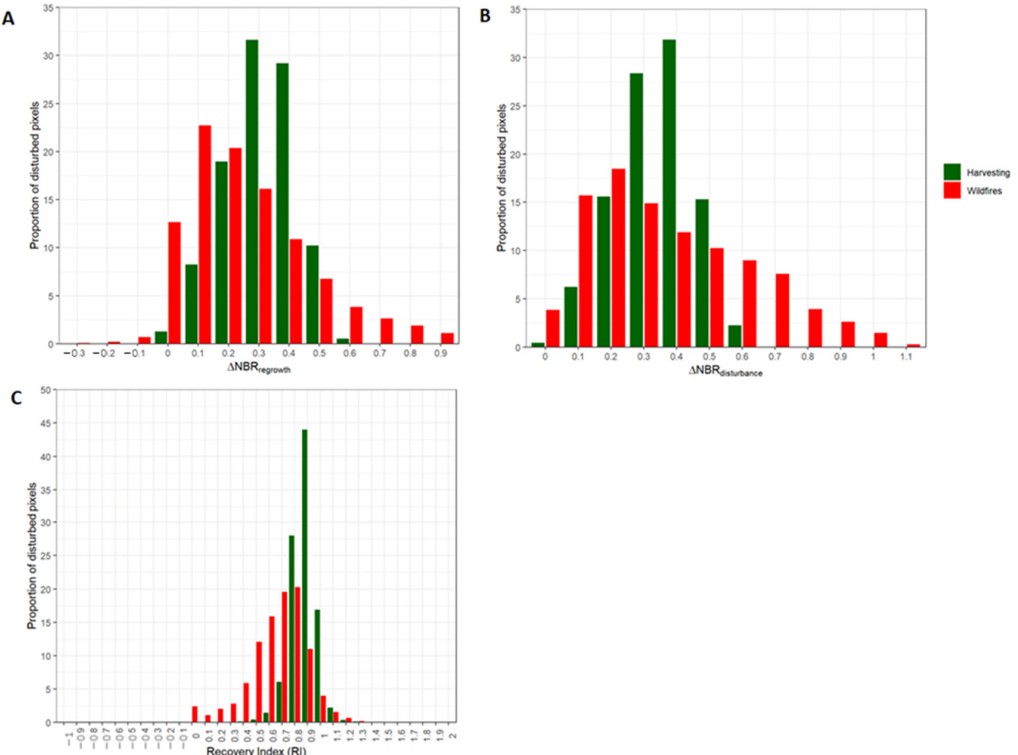

**Figure 5.** (Panel **A**)—Distribution of $\Delta NBR_{regrowth}$ for harvesting and wildfires. (Panel **B**)—Distribution of $\Delta NBR_{disturbance}$ for harvesting and wildfires. (Panel **C**)—Distribution of Recovery Index for the harvesting and wildfires.

The magnitude of the disturbance event ($\Delta NBR_{disturbance}$) showed values once again higher in absolute value for the wildfires class, with 20% of them scoring more than 0.6 in drop of NBR values in the year of disturbance, and a higher variability than the harvesting class. The average value for the wildfires class was 0.337 with a standard deviation of 0.277,

compared to the average of 0.384 and a standard deviation of 0.124 of the harvesting class (Figure 5).

Scaling the $\Delta NBR_{regrowth}$ by the magnitude of the disturbance event ($\Delta NBR_{disturbance}$), the second metric, a relative indicator of recovery (Recovery Index, RI) is obtained. The definition of RI classifies pixels with RI values $\leq 0$ as non-recovering; while all pixels of the harvesting class were recovering in the first five years after the disturbance, 11% of the wildfires class did not show signs of recovery. The same trends observed for both the $\Delta NBR_{regrowth}$ and $\Delta NBR_{disturbance}$ are maintained, with lower average and higher variability values for the wildfires class ($\mu = 0.744$, $\sigma = 0.490$) compared to the harvesting class ($\mu = 0.925$, $\sigma = 0.118$).

To evaluate the long-term aspects of recovery, the Y2R metric was calculated with three different recovery scenarios (Figure 6). Disturbances that happened before 2001 for the harvesting class could not be included in the analysis because there was not a two-year period pre-disturbance available to compute the $NBR_{pre-disturbance}$; from the 62,612 analyzed pixels, only 55,540 were included in the long-term recovery analysis.

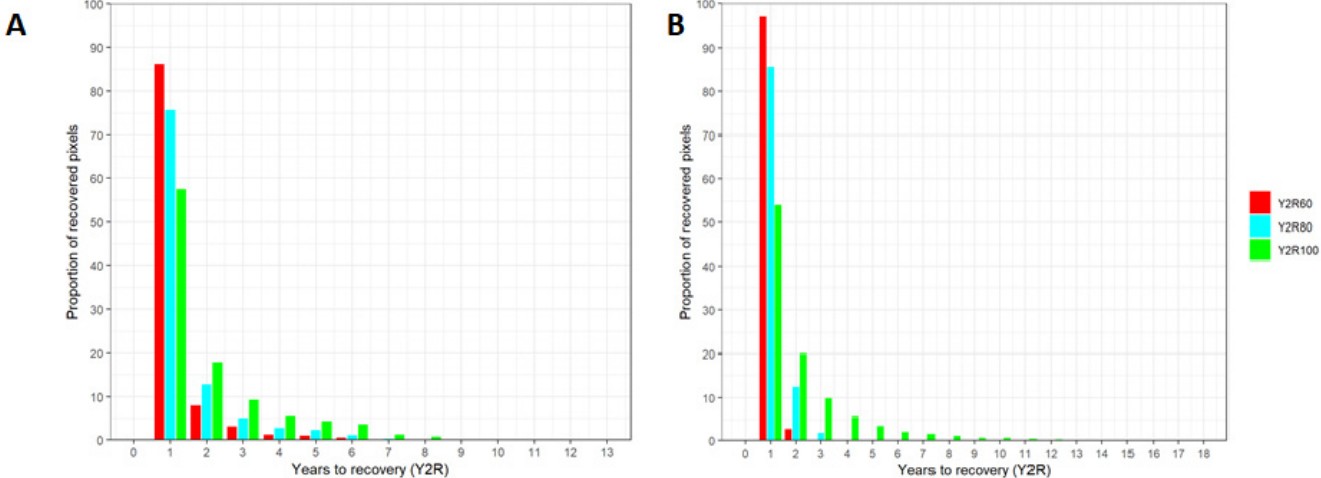

**Figure 6.** Pixel-level distribution of the different Y2R scenarios for the wildfires (Panel **A**), and for the harvesting class (Panel **B**).

For the Y2R60% scenario (Figure 6), the recovery NBR threshold of 60% of the NBR pre-disturbance value was reached after only one year from the disturbance event by both classes, with the wildfires class being the slowest, recovering approximately 87.5% of the disturbed pixels. The first year after disturbance recorded the highest proportion of pixels recovered for both classes, value that decreased in the following years.

The harvesting class observed that 100% of the disturbed pixels were recovered by the end of 2018. A complete recovery was observed for the harvesting class after only two years from the disturbance compared to the six years of the wildfires class. The average Y2R60% value for the harvesting class was 1.02 years ($\sigma = 0.16$ years) compared to 1.21 ($\sigma = 0.7$ years) for the wildfires.

For the Y2R80% scenario (Figure 6, Table 4), the threshold was reached after the first year: once again the harvesting class observed the highest recovery rate, reaching complete recovery (>99%) after only three years from the disturbance, while by 2018 only 77% of the pixels of the wildfires class reached complete recovery, even after eight years from the disturbance. The average Y2R80% value for the harvesting class was 1.1 years ($\sigma = 0.4$ years) compared to 1.4 years ($\sigma = 1.02$ years) for the wildfires class.

For the Y2R100% scenario, the threshold was still reached after the first year by both classes (Figure 6); however, complete recovery was not reached by any of them: the harvesting class scored the highest proportion of pixels recovered (48%), with the last pixels recovering after 11 years from the disturbance event; the wildfires class recovered only 33% of the total, with the last pixels recovering after eight years from disturbance. The

average Y2R100% value for the harvesting class was 2.1 years (σ = 2.02 years) compared to 1.89 years (σ = 1.50 years) for the wildfires class.

**Table 4.** Summary of the three recovery metrics; a "+" classifies the pixel for that metric as recovered, while the "−" as not recovered.

| $\Delta NBR_{regrowth}$ | RI | Y2R80% | Description | Proportion of Disturbed Pixels for the Harvesting Class | Proportion of Disturbed Pixels for the Wildfires Class |
|:---:|:---:|:---:|:---:|:---:|:---:|
| + | + | + | Recovery indicated by all 3 metrics | 99.752 | 76.510 |
| + | + | − | Short-term recovery indicated; long-term recovery not attained by 2018 | 0.072 | 22.166 |
| + | − | + | Recovery indicated by $\Delta NBR_{regrowth}$ and Y2R | 0.000 | 0.000 |
| + | − | − | Recovery indicated by $\Delta NBR_{regrowth}$ only | 0.000 | 0.000 |
| − | + | + | Recovery indicated by RI and Y2R | 0.000 | 0.000 |
| − | + | − | Recovery indicated by RI only | 0.000 | 0.000 |
| − | − | + | Long-term recovery indicated | 0.176 | 0.493 |
| − | − | − | No recovery was indicated by any of the metrics | 0.000 | 2.241 |

The highest proportion of recovered pixels according to all the metrics analyzed was scored by the harvesting class, with >99% of the pixels classified as recovered, compared to the 76% of the wildfires class; wildfires also recorded <2.5% of the pixels as not recovered by any of the three metrics.

The harvesting class had a <0.01% of pixels that did not attain long-term recovery by the end of 2018 and <0.2% that did not attain short-term recovery but was able to recover by the end of 2018. Wildfires had approximately 23% of the pixels that did not attain short-term or long-term recovery: the highest proportion, approximately 22%, did not attain long term recovery but observed short-term recovery signs; the lowest did not attain short-term recovery indicated by both short-term metrics, but was able to recover by the end of 2018, indicating a longer timespan needed to recovery with a slow increase and fast growing afterwards.

It is worth noting that when short-term recovery was attained, such a condition was indicated by both short-term recovery metrics: in our case study, there were not any cases with recovery indicated by only one of the two metrics.

*3.3. Trajectories Classification*

The averaged out-of-bag (OOB) error indicated a high accuracy of the model, showing the default value for the *ntree* parameter as optimal for the classification (Figure 7). The model had an averaged overall accuracy (OA) of 98.6%, with class errors relatively well balanced: averaged producer's accuracy (PA) scored values ranging between 94% and 99%, while averaged user's accuracy (UA) between 99.08% and 99.4%.

The mean decrease in accuracy values were averaged and used to evaluate the predictor variables importance: the mean post-disturbance metric scored as the most important predictor to differentiate between the two disturbance classes trajectories (Figure 8). Most of the recovery metrics scored extremely low on the scale, while metrics derived from spectral values or trend analysis occupy the highest part. It is worth noting that the first two most important metrics refer to post-change event conditions of the spectral trajectory.

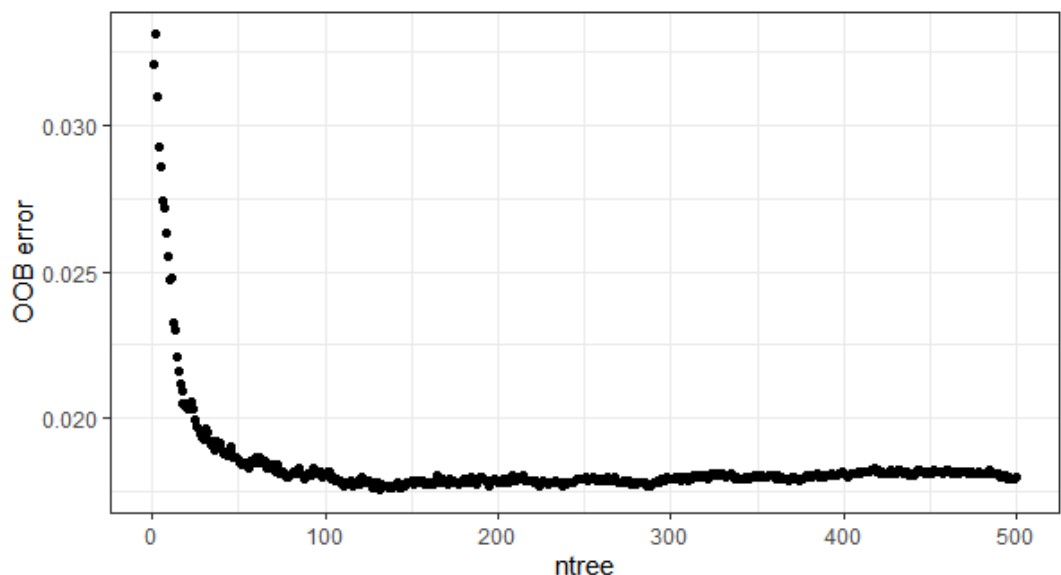

**Figure 7.** Averaged OOB error for the Random Forest model.

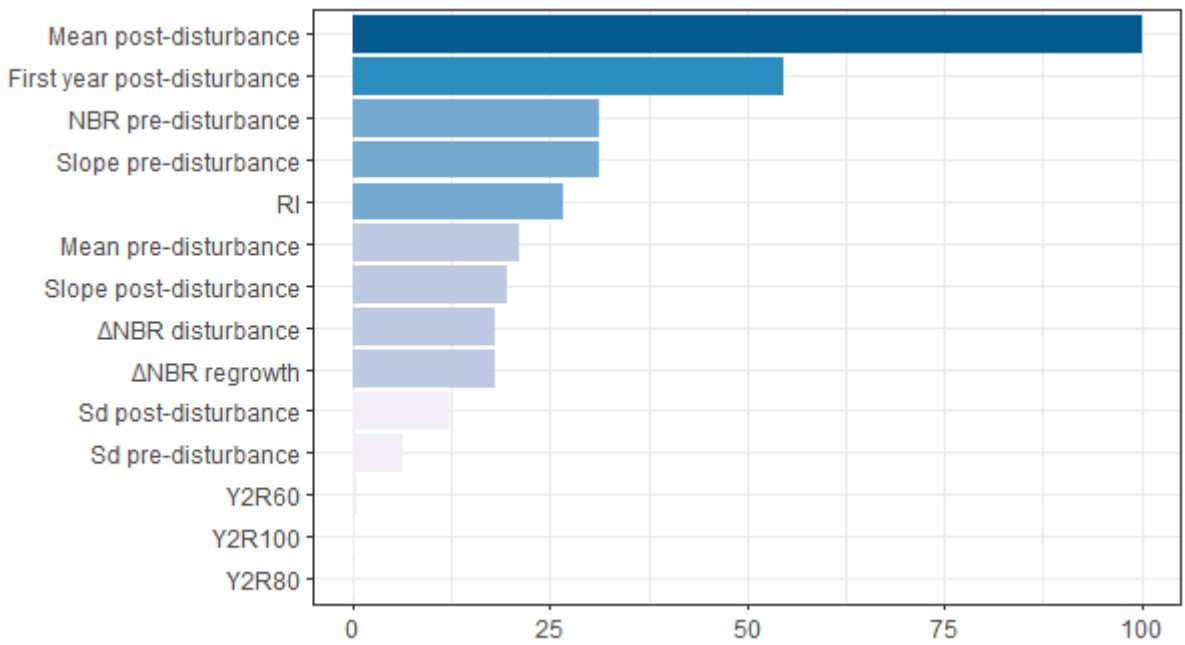

**Figure 8.** Relative variable importance for the Random Forest model.

## 4. Discussion

The aim of the study was to conduct a full temporal characterization of two disturbance agents in Mediterranean forest ecosystems. The characterization focused on the more frequent disturbances recorded for the study area, which were harvesting operations and wildfires, using a period of observation of 20 years. To provide baseline information for future disturbance monitoring and forecasting, the analysis used the widest range of spectral variables available from the Landsat archive and both short- and long-term recovery metrics used in the past to characterize similar disturbances in boreal forests.

Post-disturbance forest spectral recovery is an ongoing, continuous process described by the slow increase (indices) or decrease (bands) in the spectral variable temporal pattern. After the trajectory extraction, all spectral variables analyzed have shown a degree of sensitivity to disturbances, having different mean values before and after the year of the

disturbance event even though the disturbances analyzed and their inherent dynamics are different.

Knowledge about post-disturbance recovery capacity is fundamental information for forest management and planning. Recovery capacity represents a useful indicator for ecosystem resilience, in general, a fast recovery rate means a high level of ecosystem resilience [74]. In addition, post-disturbance recovery could influence the climate due to the effect on surface radiation balance, carbon budgets, water balance, surface albedo, soil moisture, erosion, and, finally, evapotranspiration [75]. Substantial differences in absolute change in mean values between bands and indices (Table 3) confirmed the utility of the latter over the former for change detection analysis: as hypothesized, spectral indices magnified the effects of change due to disturbance events, with change in mean values for the indices differing by one order of magnitude compared to the bands (Figures 3 and 4); singular bands remain useful to give insights on the location of disturbances in the spectro-temporal space, but spectral indices maintain the upper hand in detecting more subtle responses to disturbances and recovery from vegetation (Table 3). This study also confirmed for Mediterranean forests the major sensitivity of the Landsat NIR, SWIR 1 and SWIR 2 bands, among all bands, to detect forest disturbances [76].

The results indicate a similar temporal pattern for every spectral index (Figure 4), showing no sign of dispersion in mean values neither before nor after the disturbance, with the NBR2 index having the least inter-annual variability. This could be attributed to the fact that the NBR2 contrasts both SWIR bands and is not affected by fluctuations in the NIR or Red band like the other indices [77,78]; this information could be extremely useful to map long term aspects of disturbances in future studies [79]. The SAVI scored as the least sensitive index both to disturbance detection and recovery, the MSAVI and EVI reached values close to the NDVI: this reflects what was found by Storey et al. [78] in Mediterranean-type ecosystems dominated by shrublands formations (*chaparral*) in California. However, differences for all the indices found in this study are less enhanced and the reason could be attributable to vegetation density: *chaparral* vegetation is generally more open than sclerophyll forests and maquis formations of the Mediterranean Basin. It can be assumed that soil-adjusted indices gain significance in change detection analysis the denser the forests formations analyzed are.

Greenness indices (EVI, NDVI, SAVI, and MSAVI) addressed disturbances correctly but failed to portray post-disturbances conditions accordingly: wetness indices (NBR, NBR2, NDMI) patterns remarked that mean values similar to pre-disturbance conditions take some years, from three to 10 depending on the class of disturbance, to be reached; instead, all greenness indices displayed pre-disturbance mean values, and in some cases even higher, after only two or three years for every class of disturbance. This suggests a fast regain in photosynthetic activity after the disturbance that could be attributable to colonizing grasses and shrubs in clearings and epicormic shoots, but also as fast regrowth after coppicing as showed by Chirici et al. [35], being that the coppice system is the most common form of forest management in our study area.

The NBR was found to be the most sensitive index to capture disturbances, which aligns with the results obtained by Chirici et al. [35] and Giannetti et al. [27] in Mediterranean forests, and by Kennedy et al. [57] in boreal forests. However, an in-depth analysis of our results on the recovery metrics pointed out a difference between the two biomes (i.e., boral and Mediterranean forests) in magnitude of disturbance events, regrowth pattern, and rate of regrowth. In fact, the recovery of Mediterranean forests is faster than the ones observed by Kennedy et al. [57] and White et al. [80], confirming the results shown by Chirici et al. [35], Giannetti et al. [27], and Francini et al. [32].

The results of short-term recovery metrics showed that harvesting disturbances are on average of medium intensity (85% of the pixels had a $\Delta NBR_{disturbance}$ value comprised between 0.2 and 0.5) with scarce variability, meaning that harvesting operations had regular intensity on all forest types across all 20 years. They also indicated a high rate of recovery, with the long-term metric (Y2R80%) assessing only <0.1% of pixels recovering

after five years from the disturbance. That could be expected since as with natural disturbances, forest harvesting varies in frequency, distribution, and intensity but differs from them because it is essentially driven by socioeconomic factors and human decisions [81], especially in privately owned properties. Harvesting simulates effects produced by natural disturbances [82] and management systems try to do so to sustain forest dynamics and biodiversity while still allowing harvesting operations.

In some situations, harvesting practice could create less damage to forest vegetation with respect to the action of natural disturbance agents [83]. Traditionally harvesting operations are realized on the most productive sites where the combined force of forest management and site characteristics allows harvest areas to return to the forest state in a brief period of time. These sites are strongly influenced by humans, which have strongly altered the natural fire regime and excluded fire from these ecosystems. Human actions have created a dense forest with lower structural variation and complexity that are ideal for the development of megafire [83–85]. In this sense, the reintroduction of the natural fire regime could be particularly useful in improving the ecological variability in forests [85].

This is especially true for coppice stands, where, after cutting, plants and shrublands grow between the stools and cover the open area, as confirmed by LiDAR analysis done by Chirici et al. [35], left by and, after five or so years, the increasing shade made by the closing coppice canopy rapidly eliminates most of the foliage beneath. There are no further changes in the forest structure until the next cut, apart from the growing coppice stems.

Overall, information derived from recovery metrics and the classification model shows that harvested areas recovered faster in both short-and long-term aspects and showed less variability in magnitude of the disturbance event and rate of recovery over time than wildfires. Even though recovery metrics did not score very high on the feature importance scale, the model identified a different behavior in the spectral pattern between disturbances based on the spectral values recorded in the recovery segment of the trajectories, from which the recovery metrics are derived. Taking the averaged NBR trend and the first two most important metric selected by the model as an example, there is a huge relative gap between mean post-disturbance and first year post-disturbance values for the two classes: higher values and a faster recovery rate for harvesting mean a different collocation of the trend in the spectro-temporal space, which allows clearly distinguishing between disturbances.

Fire intensity determines the effects of fire on vegetation: low intensity events usually leave residual vegetation on site in the form of standing snags or living trees, high intensity events can kill all living biomass; both influence regeneration processes, with residual vegetation offering protection to new saplings and seedlings, while high intensity fires can release soil nutrients to the soil, making them readily available to the soil seed bank, or help pyrophyte species seeds to germinate.

Short term recovery metrics indicated a great variability in the magnitude of observed fire events, with very different recovery rates: all pixels that suffered high intensity events ($\Delta NBR_{disturbance} \geq 0.8$) also displayed a high positive recovery rate (RI > 0.8 on average) and belonged to only three of the eleven forest types analyzed (Downy oak, Maquis formations, and Maritime pine); instead, pixels interested by low intensity events showed a wider range of behaviors, with fast recovery (RI > 1) or no recovery at all (RI < 0).

The long-term recovery metric indicated a longer period needed to achieve full recovery than harvested areas (1.4 years against 1.1 on average); pixels recovered in the first year were interested by low intensity events ($\Delta NBR_{disturbance} \leq 0.2$ on average), while all pixels interested by high intensity events, despite the displayed high recovery rate, achieved full recovery only after four years on average. However, also if fire damaged areas need in average more time to recover, compared to harvested areas, it is important to remember that most of the forest tree species present in the fire damaged areas are well adapted to fire.

These results lead to a better understanding of the dynamics of fire events in Mediterranean forests: fire events have higher variability in intensity and area coverage (the largest observed in this study covered more than 300 hectares) and need a longer period to achieve full recovery (up to nine years) than harvested areas. High intensity fire events are less

likely to occur than low intensity ones and interest only certain forest types that are well-known for being adapted to fire or require fire to germinate [83]: this interpretation explains the high recovery rates found for pixels with high $\Delta NBR_{disturbance}$ values. The exception is the Downy oak, a species with poor resprouting capabilities, but the understory vegetation of these stands is composed of the same pyrophyte species found in the maquis formations [84]; thus, the high recovery rates observed are attributed to the understory component and not to the main species of the forest type. It is important to remember that for forest fire damaged areas from satellite, we can just understand the recovery of photosynthetic activities, and it is not possible to distinguish if the species responsible for the recovery are the previous forest tree species or invasive species. However, suppose we have a recovery of photosynthetic activities, even invasive species can have a beneficial effect on the area: for example, they can contest the erosion of the slope and start a secondary succession in the area.

## 5. Conclusions

This study contributed by outlining the differences among two classes of disturbance in Mediterranean forest ecosystems through the information derived by Landsat spectral trajectory analysis based on bands and vegetation indices and NBR based spectral, trend analysis, and recovery metrics. The obtained results offer opportunities for future studies in multiple directions to understand disturbance phenomena and recovery processes, including the creation and application of new NBR based recovery metrics adapted to the significative shorter temporal recovery domain of Mediterranean areas compared to boreal forests, a more accurate characterization of the disturbances using longer, more dense, and intra-annual LTS in Mediterranean areas, or the characterization of other agents of disturbance with patterns not clearly documented. With these pieces of information, the development of approaches for automating the attribution of disturbance type could not be far. Future studies need to take into consideration also other types of abiotic and biotic disturbances, such as windthrow and insect disturbances, that were considered in this study since we do not have a reference dataset useful for trajectory analysis.

**Author Contributions:** Conceptualization, G.C. and F.G.; methodology, C.B.; formal analysis, C.B.; photointerpretation: E.V. and G.D.; writing—original draft preparation, C.B. and F.G.; writing—review and editing, C.B., M.P., F.G., D.T., G.C., E.V. and G.D. All authors have read and agreed to the published version of the manuscript.

**Funding:** This research was funded by GO-SURF Decision support to sustainable forest planning, Tuscany Rural development 2014–2020 for Operational Groups (in the sense of Art 56 of Reg.1305/2013).

**Institutional Review Board Statement:** Not applicable.

**Informed Consent Statement:** Not applicable.

**Data Availability Statement:** The data presented in this study are available on request from the corresponding author.

**Conflicts of Interest:** The authors declare no conflict of interest.

## Appendix A

**Table A1.** Specification of the Landsat images used in the study.

| Satellite | Sensor | Processing Level | WRS2 Address | Acquisition Date | Collection | Tier | Product |
|-----------|--------|------------------|--------------|------------------|------------|------|---------|
| Landsat 5 | TM | L1TP | 192/030 | 26 June 1999 | 01 | T1 | sr |
| Landsat 5 | TM | L1TP | 192/030 | 15 August 2000 | 01 | T1 | sr |
| Landsat 5 | TM | L1TP | 192/030 | 2 August 2001 | 01 | T1 | sr |
| Landsat 5 | TM | L1TP | 192/030 | 18 June 2002 | 01 | T1 | sr |

**Table A1.** *Cont.*

| Satellite | Sensor | Processing Level | WRS2 Address | Acquisition Date | Collection | Tier | Product |
|-----------|--------|------------------|--------------|------------------|------------|------|---------|
| Landsat 5 | TM | L1TP | 192/030 | 8 August 2003 | 01 | T1 | sr |
| Landsat 5 | TM | L1TP | 192/030 | 7 June 2004 | 01 | T1 | sr |
| Landsat 5 | TM | L1TP | 192/030 | 26 June 2005 | 01 | T1 | sr |
| Landsat 5 | TM | L1TP | 192/030 | 13 June 2006 | 01 | T1 | sr |
| Landsat 5 | TM | L1TP | 192/030 | 18 July 2007 | 01 | T1 | sr |
| Landsat 5 | TM | L1TP | 192/030 | 21 August 2008 | 01 | T1 | sr |
| Landsat 5 | TM | L1TP | 192/030 | 23 July 2009 | 01 | T1 | sr |
| Landsat 5 | TM | L1TP | 192/030 | 10 July 2010 | 01 | T1 | sr |
| Landsat 5 | TM | L1TP | 192/030 | 27 June 2011 | 01 | T1 | sr |
| Landsat 7 | ETM+ | L1TP | 192/030 | 8 August 2012 | 01 | T1 | sr |
| Landsat 8 | OLI/TIRS | L1TP | 192/030 | 16 June 2013 | 01 | T1 | sr |
| Landsat 8 | OLI/TIRS | L1TP | 192/030 | 6 August 2014 | 01 | T1 | sr |
| Landsat 8 | OLI/TIRS | L1TP | 192/030 | 6 June 2015 | 01 | T1 | sr |
| Landsat 8 | OLI/TIRS | L1TP | 192/030 | 27 August 2016 | 01 | T1 | sr |
| Landsat 8 | OLI/TIRS | L1TP | 192/030 | 14 August 2017 | 01 | T1 | sr |
| Landsat 8 | OLI/TIRS | L1TP | 192/030 | 17 August 2018 | 01 | T1 | sr |

## Appendix B

**Table A2.** Landsat spectral indices used in the study.

| Index Type | Spectral Index | Formula Used by USGS Processing |
|------------|----------------|--------------------------------|
| Greenness | Enhanced Vegetation Index (EVI) | $G \times \frac{(NIR-Red)}{(NIR+C1 \times Red - C2 \times Blue + L)}$ <br> Where: <br> $G = 2.5$ <br> $C1 = 6$ <br> $C2 = 7.5$ <br> $L = 1$ |
| Greenness | Normalized Difference Vegetation Index (NDVI) | $\frac{NIR-Red}{NIR+Red}$ |
| Greenness | Modified Soil Adjusted Vegetation Index (MSAVI) | $\frac{(2 \times NIR + 1 - \sqrt{(2 \times NIR + 1)^2 - 8 \times (NIR - Red)})}{2}$ |
| Greenness | Soil Adjusted Vegetation Index (SAVI) | $\frac{(NIR-Red)}{(NIR+Red+L)} \times (1+L)$ <br> Where: <br> $L = 0.5$ |
| Wetness | Normalized Burned Ratio (NBR) | $\frac{NIR-SWIR\,2}{NIR+SWIR\,2}$ |
| Wetness | Normalized Burned Ratio 2 (NBR2) | $\frac{SWIR\,1-SWIR\,2}{SWIR\,1+SWIR\,2}$ |
| Wetness | Normalized Difference Moisture Index (NDMI) | $\frac{NIR-SWIR\,1}{NIR+SWIR\,1}$ |

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
