# Peer review of "Characterization of Wildfires and Harvesting Forest Disturbances and Recovery Using Landsat Time Series: A Case Study in Mediterranean Forests in Central Italy"

_fire, doi:10.3390/fire5030068_

Round 1
Reviewer 1 Report
This study used time series Landsat data to characterize the harvest and fire disturbances in Mediterranean forests. The work is interesting and fundamental for forest ecosystem monitoring. However, a major revision is necessary before it can be published in Fire. The method, result and discussion sections need better structured. The current version is a little difficult to follow and some descriptions are confused. Some specific comments are as follows:
- Line 23 short and long-term aspects was not clearly defined in the paper.
- Line 24 as I understand, only the metrics from NBR were used? spectral metrics is confused.
- Line 33. I don’t think the keywords are proper according the paper. Fire, forest, forest fires, clearcut can be changed to harvest, wildfire, forest?
- The introduction section should be better organized. The current version is very fragmented.
- Line 122 format error.
- Line 139 Figure 1 lack of harvest areas?
- Line 161 I did not see much analysis about the forest types except in the discussion section.
- Line 170 is 8.545 correct?
- The three reference data should all mapped in Figure 1.
- It is better to give an example to show how the outliers were removed.
- How did you process the difference between the recorded change year and detected change year in Jump shape from ShapeSelectForest?
- Line 230 the metrics should be clearly depicted or defined.
- Line 257 where was Table 2 referred in the text?
- How did you define the ΔNBR pre-disturbance by the “Arithmetic mean of the first two years before the change event”? It is confused.
- The metrics for NBR and bands and VIS can all be summarized in Table 2. The short and long term can be clearly defined here.
- Line 265 according to the results, you only used the metrics from NBR in the classification model, Am I right? So it is better to clearly stated here and not refer to 2.5. because 2.5 was method for all bands, indices.
- Standard deviation in Figure 2 is necessary.
- Line 326 is hard to understand.
- The ΔNBRregrowth and ΔNBRdisturbance were not defined in the method section, or not consistent with Table 2.
- Could Figure 4 C be consistent with A and B?
- Line 341 RI should be introduced in the method section.
- Line 417 the spectral values is confused as you just used NBR, not bands.
- The discussion section should be more focused on you results. You mentioned the forest types data in the method section, however, nothing was analyzed in the result section. The discussions lack of such results. It will be interesting if you find more support of the different change trajectories or characteristics for different forest types.
- The discussion section should be better organized. The current version is hard to follow.
Author Response
Reviewer #1 (anonymous)
This study used time series Landsat data to characterize the harvest and fire disturbances in Mediterranean forests. The work is interesting and fundamental for forest ecosystem monitoring. However, a major revision is necessary before it can be published in Fire. The method, result and discussion sections need better structured. The current version is a little difficult to follow and some descriptions are confused.
Dear Review, thanks a lot for your suggestion that we think improved a lot of our manuscript. We try to reply to all your comments and we follow all your suggestions that we think improved the manuscript a lot. Thank again for your work!
The number of the lines that we reported in the reply are refer to the clear version and not the track change.
Some specific comments are as follows:
Line 23 short and long-term aspects was not clearly defined in the paper.
RE: we defined short and long term forest disturbances please see line 112-113. Moreover in order to clarify better the recovery metrics we added a new figure 2 that explain better what they means.
Line 24 as I understand, only the metrics from NBR were used? spectral metrics is confused.
RE: Yes, in the end we just used the metrics defined in Table 2. The words “recovery” and “trend analysis” metrics already comprise the definition of “spectral”. We removed the word “spectral” for this reason as it is indeed redundant.
Line 33. I don’t think the keywords are proper according the paper. Fire, forest, forest fires, clearcut can be changed to harvest, wildfire, forest?
RE: we changed it according to your suggestion
The introduction section should be better organized. The current version is very fragmented.
RE: we reorganized it in accordance also with the other suggestion of review n. 2
Line 122 format error.
RE: yes we changed it
Line 139 Figure 1 lack of harvest areas?
RE: Yes, this was a mistake! We added the harvested areas in Figure 1.
Line 161 I did not see much analysis about the forest types except in the discussion section.
RE: See following comment on the discussion section: sample is too limited, especially for some forest types. It would not have been significant to draw general conclusions from such a small sample when the distribution of pixels per disturbance and forest type is not even. Aggregated values (i.e. all forest types) can give a general indication of how Mediterranean forests react to those disturbances, but for more detailed (i.e. per forest type) conclusions more research is needed.
Line 170 is 8.545 correct?
RE: It is a formatting error, should have been 8,545 ha. It is now fixed (see line 199)
It is better to give an example to show how the outliers were removed.
RE: See following comment on the discussion section:
How did you process the difference between the recorded change year and detected change year in Jump shape from ShapeSelectForest?
RE: We’re not sure if we understand the question. The package fits a function according to the specific shape decided and does not detected a change on its own: the difference between the change year (y = 0) in the recorded change (raw values) and the change in the trend (fitted values) is only in reflectance/band ratio values (y axis in the plots) not in time (x axis).
Line 230 the metrics should be clearly depicted or defined.
RE: Metrics are later defined in Table 2. To make line 313, Table 2 and section 2.6 more clear we decided to add an additional figure showing all NBR recovery metrics on the NBR trend.
Line 257 where was Table 2 referred in the text?
RE: Table 2 is now referenced in the text in section 2.6, see line 268; previous absence of this reference was due to formatting error.
How did you define the ΔNBR pre-disturbance by the “Arithmetic mean of the first two years before the change event”? It is confused.
RE: See now the new figure 2 with the formulation of the recovery metrics. Some of them (Mean, standard deviation and slope pre- and post- disturbance) are not described in the figure due to their obvious formulation.
The metrics for NBR and bands and VIS can all be summarized in Table 2. The short and long term can be clearly defined here.
RE: See previous comment, short and long term are now defined at line 112-113 and in figure 2 we clarify the definition of the metrics.
Line 265 according to the results, you only used the metrics from NBR in the classification model, Am I right? So it is better to clearly stated here and not refer to 2.5. because 2.5 was method for all bands, indices.
RE: That’s correct, it was a mistake during the editing of the manuscript. Reference should have been not 2.5 but 2.6, where the metrics are introduced together with references and formulations (see Table 2, Figure 2)
Standard deviation in Figure 2 is necessary.
RE: We added the standard deviation of the aggregated raw values to the plots see figure 3 and 4. We initially decided to not include it since the figure looks more difficult to read now, even though the main trends (i.e. which band/spectral index is more informative) can still be picked up.
Line 326 is hard to understand.
RE: We rephrased it to make it more understandable.
The ΔNBRregrowth and ΔNBRdisturbance were not defined in the method section, or not consistent with Table 2.
RE: We added the missing metric to Table 2
Could Figure 4 C be consistent with A and B?
RE: The time is not the same for the pannel B and C because the data of the fires do not cover all the time of the analysis of the harvesting area it is clear mentioned in the description of the database. So we can not be consistent with the x axis.
Line 341 RI should be introduced in the method section.
RE: We followed the suggestion and added the RI to the method section in Table 2. References for this metric are the same of the other metrics, see the following:
White, J.C.; Wulder, M.A.; Hermosilla, T.; Coops, N.C.; Hobart, G.W. A nationwide annual characterization of 25 years of forest disturbance and recovery for Canada using Landsat time series. Remote Sens. Environ. 2017, 194, 303–321, doi:10.1016/j.rse.2017.03.035
Moreover we added the new figure 2 to be more clear in the description of the metrics
Line 417 the spectral values is confused as you just used NBR, not bands.
RE: That is correct, only NBR was used: we fixed the sentence in a way that is more consistent with the other definitions we gave in the manuscript.
The discussion section should be more focused on you results. You mentioned the forest types data in the method section, however, nothing was analyzed in the result section. The discussions lack of such results. It will be interesting if you find more support of the different change trajectories or characteristics for different forest types. The discussion section should be better organized. The current version is hard to follow.
RE: We did not follow through with discussing results by forest type since the sample is too limited, especially for some forest types. It would not have been significant to draw general conclusions from such a small sample. For this reason we followed through interpretations of the results that are general (i.e. for all forest types) but did not delve into more specific (i.e. per forest type) interpretations/speculations of our results. We agree though that it is an interesting topic to pursue/follow through: the assumption in this case would be that each forest type would react differently to disturbances, with some types recovering sooner or later than others and in different ways across a wide range of agents of disturbance. More research in this direction is certainly needed.

Reviewer 2 Report
Good job with the analysis and the manuscript. It is an interesting topic and in general well-written. Please find my suggestions in the attached pdf. There are areas that need improvement, especially around the "fire".
- My major concern in this analysis is around how you define recovery following a disturbance event. Recovery in "spectral reflectance" can be different from recovery on the ground in an ecologically sensible way. How can you distinguish recovery by different vegetation or even invasive species, from your target vegetation?.
- Be more explicit about how you define "fire" in this landscape. Is it a fire-adapted ecosystem, with fire being a natural phenomena? If so, what's the current fire regime, how altered is it, and is it fine for these systems to take additional time to recover.. if it's ecologically that's what it needs. Please explain!
I think this is an interesting study. Good luck!
Thank you.

Author Response
Reviewer #2 (anonymous)
Good job with the analysis and the manuscript. It is an interesting topic and in general well-written. Please find my suggestions in the attached pdf. There are areas that need improvement, especially around the "fire".
Dear Review, thanks a lot for your suggestion that we think improved a lot of our manuscript. We try to reply to all your comments and we follow all your suggestions that we think improved the manuscript a lot. Thank again for your work!
The number of the lines that we reported in the reply are refer to the clear version and not the track change.
My major concern in this analysis is around how you define recovery following a disturbance event. Recovery in "spectral reflectance" can be different from recovery on the ground in an ecologically sensible way. How can you distinguish recovery by different vegetation or even invasive species, from your target vegetation?
RE: Thanks for your comment! Yes for fires we do not have evidence regarding if the area are recovery by the previous forest tree species or if they were recover by invasive species since from satellite we cannot distinguish that. However, also if the invasive species will be in the responsible for the recover that will be important for contrast for example erosion of the slopes. However, for harvesting area we know, also from the results of previous study, that the recovery is reach by the same species since the harvested area are mostly in coppices forest. We add some sentences in the discussion regarding that aspect please see line 541-545 and line 588-594
Be more explicit about how you define "fire" in this landscape. Is it a fire-adapted ecosystem, with fire being a natural phenomena? If so, what's the current fire regime, how altered is it, and is it fine for these systems to take additional time to recover.. if it's ecologically that's what it needs. Please explain!
RE: we add in the introduction section more information on the fire regimes in our study area so you can have more information at line 43-59. Moreover see our previous reply.
More specific comments from R2 are in the following document: link
I think the authors have done a good job giving a good introduction to the remote sensing methodological approach discussed in this paper. However, I think it would be better if you have a paragraph giving an intro to the two types of disturbances discussed here. Harvesting - how large is the industry here, what type of a landscape-scale impact does harvesting operations have etc.. Then fire - are those mostly uncontrolled wildfires, is this area historically prone to fires, is it a natural phenomena in this region where most tree species/vegetation are fire adapted? Etc..
RE: we add a new paragraph regarding that aspect please see line 43-59. We added a description of the two types of disturbances and their importance in the Mediterranean area with particular details for Italy.
advantages over what? ground sampling?.. maybe better to say "..tools are widely used mainly due to these six reasons"
RE: we rephrase the sentences please see line 55-59
combine these two paragraphs into a single para.
RE: DONE see line 77-79
replace by "and"
RE: DONE
combine "twenty-year"
RE: DONE
linking back to my previous comment on the introduction.. tell us about the fire history in this region.. is fire a natural phenomena, and the oaks and pine species are fire adapted?..
RE: Yes we agreed. We added this information in the section related to the study area description. We reported the details related with the adaptation of the different species to fire, the possible risk. Please see line 149-153 and line 159-163.
maybe "undisturbed" is a better replacement - throughout!
RE: to be consistent within all the manuscript we change not-distrubed with undisturbed in all the manuscript and figures
spelling mistake “wheter”
RE: DONE
There were few error messages in the manuscript. Please address them
RE: We do not know why it happened. We changed all the errors in the manuscript with the right references.
no need to capitalize - check through out!
RE: changed thanks for the suggestion
define the first time you have mentioned this OBB
RE: we define the terms out-of-bag error
Please see my comments below - also, please consider having a paragraph discussing how important this information could be from a forest management perspective
RE: we add a small paragraph 532-540
Not sure what this means, and I cannot agree this to be a generalized statement. Maybe this is true for some areas, but not everywhere. "Fire" is a natural disturbance - and you cannot say it damages an environment if the fire happens in a fire adapted ecosystem. This issue is due to anthropogenic influences, the fire regimes in many places has changed and the with the current vegetation after decades of fire suppression (similar to what you see in many fire adapted ecosystems in the US), the natural fire occurrence becomes mega wildfires causing much damage.. in those places the objective should be to re-introduce fire to the landscape, and re-instate the natural fire regime gradually. In such fire adapted systems fire is needed to maintain the ecological function and biodiversity... I would like to see this discussion in this section (with citation - you would find many).
RE: we rewrite in this sense the paragraph see also our previous comment
This is another question I had - the results show that the harvested areas recover in 2-3 yrs. How can you be sure if this recovery is attributable to the development of the coppice canopy? Couldn't it be any other shrubs, vegetation, or even invasive species? We know that open areas are prone to invasive species expansion..? Do you have evidence to say in the analysis that the recovery is indeed a recovery of favorable vegetation..
RE: Yes we have evidence, because in some areas we have LIDAR data and we calculate the height of vegetation regrowth. We added the reference to Chirici et al. (2020) and we discussed these aspects better. Thanks a lot for your suggestion.
This is important information - since this paper is being published in the "Fire" journal, it's likely that the readers are more informed and interested on fire effects and the role of fire. As I have mentioned above, please have this information included in the introduction and M&M sections - please expand this to explain where the fire adapted landscapes are in this region, main species associated with fire.. and the natural fire regime, how altered they are.. Etc.
RE: we added more details in the introduction, material and methods, in the introduction and in the discussion related with that. Please see

Round 2
Reviewer 1 Report
The manuscript was well revised and all the questions were responsed. I think it can be published in Fire. There are some format errors that should be corrected, such as space line 241; the fonts in the Figures should be clearer and have same size among all the figures; references format should be identical.
Author Response
Dear Review,
thanks for your comments and for the help in the improving of our manuscript.
We read again it and we made all the changes that you ask us.
In particular we corrected the typos and the fonts in the Figures.
Thanks again
Reviewer 2 Report
Thank you for addressing the concerns raised during the initial review and for responding to the suggestions made. You have done a good job revising the manuscript, and I think the quality of the paper has increased with the new information you have provided with much more clarity.
There were a couple of spelling mistakes that I noticed, so give it another good read. Apart from that the only edit I suggest is on line 149, where it would be better to say "low-intensity" wildfires.
Good luck with the rest of the process!
Author Response
Dear Review,
thanks for your comments and for the help in the improving of our manuscript.
We read again it and we correct all the typos and the fonts in the Figures according also to the request of reveiw n.1.
Thanks again!